# Delivering Chemotherapy to a Metastatic Poor Risk Testicular Cancer Patient on Hemodialysis

**Kieran J. Moore** [1]**, Stephanie Snow** [2] **and Lori A. Wood** [1,2,*]

1   Department of Urology, Queen Elizabeth II Health Sciences Center, Dalhousie University,
    Halifax, NS 6299, Canada; kieran.moore@nshealth.ca
2   Department of Medicine, Queen Elizabeth II Health Sciences Center, Dalhousie University,
    Halifax, NS 6299, Canada; stephanie.snow@nshealth.ca
*   Correspondence: lori.wood@nshealth.ca; Tel.: +1-902-473-5469

**Abstract:** A standard curative intent approach of chemotherapy treatment for metastatic testicular cancer has been well established. However, there is little guidance for patients undergoing hemodialysis (HD) who require chemotherapy for this disease. Thus, we describe our treatment approach and rationale for a patient on HD with poor risk metastatic nonseminomatous germ cell tumor involving the testicle, lymph nodes, liver, and bone. After orchiectomy, five cycles of cisplatin and modified dose etoposide were delivered and strategically timed with HD. Treatment was complicated by significant neuropathy. Surgical resection of two liver lesions was performed after chemotherapy. Ten years post-chemotherapy, he remains free of clinical, biochemical, or radiological recurrence. While our patient remains free of disease after this treatment, the optimal chemotherapy and dialysis dose and schedule to maximize cure and minimize toxicity remains unknown.

**Keywords:** testicular cancer; chemotherapy delivery; hemodialysis; renal failure

## 1. Introduction

The goal of treatment for patients with metastatic testicular cancer is cure. The standard chemotherapy for treatment of poor risk patients is four cycles of bleomycin, etoposide, and cisplatin (BEP) or etoposide, ifosfamide, and cisplatin (VIP) [1]. For testicular cancer patients with renal impairment requiring hemodialysis (HD), the optimal delivery and efficacy of chemotherapy is a rare and challenging clinical scenario. We present a case of a patient with poor risk nonseminomatous germ cell tumour in the setting of chronic renal failure who was being worked up for a renal transplant. Very little literature has been published on the safe and effective delivery of chemotherapy in this subset of patients; thus, this case report provides an important addition to medical literature.

## 2. Case Report

A 30-year-old patient awaiting a living-related donor renal transplant for chronic renal failure (creatinine 571 umol/L) related to reflux nephropathy presented to the local emergency room with right upper quadrant pain. Laboratory investigations revealed elevated liver function tests with hepatomegaly on abdominal ultrasound and multiple hepatic masses suspicious for metastasis. Subsequent computed tomography of the chest, abdomen, and pelvis confirmed multifocal hepatic lesions (Figure 1), periportal lymph nodes, a 4-mm nonspecific left lower lobe lung lesion, sclerotic lesions of the ribs, and a large mass in the left scrotum consistent with a testicular cancer (Figure 2). Bone scan also demonstrated sclerotic lesions in the seventh thoracic vertebrae, sternum, and the left iliac bone. Physical exam confirmed a large right testicular mass. His alpha fetal protein (AFP) was 62,777 umol/L (normal < 10), Beta-human chorionic gonadotropin (B-HCG) was 32 IU/L (normal < 1), and the lactic acid dehydrogenase (LDH) was 367 U/L (normal < 192).

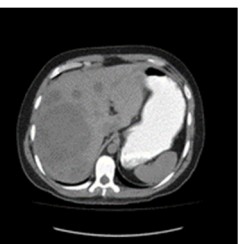

**Figure 1.** Baseline unenhanced CT scan showing multiple liver metastases.

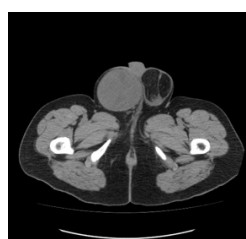

**Figure 2.** Baseline unenhanced CT scan showing a large right scrotal mass.

Orchiectomy was performed the following day and revealed a diffuse $16 \times 8 \times 6.5$ cm multi-lobulated variegated tumor mass with multifocal areas of necrosis and cystic components with no normal testicular parenchyma seen. Microscopic exam displayed predominantly immature teratoma with foci of yolk sac tumor involving the tunica vaginalis and spermatic cord.

To summarize, he had a T3N0M1bS3 tumor and thus was poor risk based on the International Germ Cell Cancer Collaborative Group criteria [2]. At this point, the patient had not yet required dialysis for his significant renal dysfunction.

The patient was started on chemotherapy as an inpatient consisting of full dose Cisplatin 20 mg/m$^2$ and Etoposide 60 mg/m$^2$ (60% dose) days 1 to 5 every 21 days. It was clear that to deliver cisplatin-based chemotherapy safely, he would need to go on HD. He was hemodialyzed one hour post-chemotherapy for days 1 to 6 for cycle 1 and days 1 to 5 for subsequent cycles. Daily G-CSF 480 mcg was administered after each cycle subcutaneously starting 24 h after chemotherapy for a total of 10 days. He was hemodialyzed three times per week when not receiving chemotherapy. He did not receive bleomycin.

Despite G-CSF, he developed febrile neutropenia on cycle 1 day 10. The absolute neutrophil count remained $<1.0 \times 10^9$/L for 5 days, and his urine cultures were positive. The dose of etoposide was therefore not increased for subsequent cycles as planned. After four cycles of chemotherapy, he had grade 2 anorexia, weight loss (16%), and grade 1 sensory neuropathy. Tumor marker levels declined during chemotherapy, and after four cycles, his AFP, B-HCG, and LDH levels were normal.

Given the dose reduction in etoposide during cycles 1 to 4 (total dose of 1200 mg instead of 2000 mg) and omission of bleomycin and ifosfamide, it was decided to proceed with a fifth cycle of chemotherapy. His neuropathy worsened (grade 2) after the fifth cycle, and no further chemotherapy was given. Three months after chemotherapy terminated, he had developed grade 3 neuropathy, both sensory and motor, primarily affecting his upper extremities. Functionally, he had significant impairment and required assistance in activities of daily living such as dressing, eating, bathing, and mobilizing. Electromyography studies showed a diffuse peripheral sensory neuropathy with mild motor involvement.

Nineteen months from diagnosis, the largest liver lesion demonstrated growth on imaging; however, tumor markers remained normal (Figure 3). Resection of this large lesion in segment 7, as well as a wedge resection of a smaller lesion in segment 8, was performed. Pathology revealed mature teratoma in both the 5.5 cm and 0.8 cm lesions.

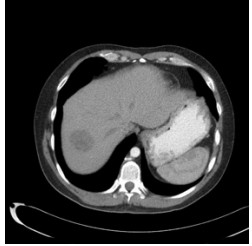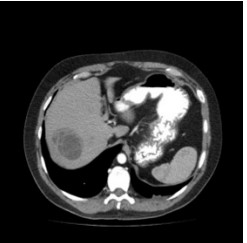

**Figure 3.** From left to right, Post-chemotherapy unenhanced CT Scan showing improvement in liver metastases and 18 month follow up enhanced CT scan showing enlarging dominant liver metastases.

His neuropathy gradually improved over the years; however, he still requires a walker for longer distances and an electric wheelchair during the icy winter months. He continues to require assistance with fine motor skills such as buttoning, tying laces, and zippers. He has not been able to work since his testicular cancer diagnosis due his dialysis schedule and the neuropathy. Ten years after finishing chemotherapy, he remains in follow up, with no evidence of clinical, biochemical, or radiologic recurrence. The bone lesions regressed over time. He remains on regular HD and has opted not to proceed with a kidney transplant.

## 3. Discussion

The metabolism of many chemotherapeutic agents is dependent on adequate renal function, which makes management challenging for patients with significant renal dysfunction requiring HD. The literature on optimally prescribing cytotoxic drugs in patients on HD is scant and often conflicting. Cisplatin is predominantly (90%) eliminated by the kidneys and present in both plasma-bound and free forms, with the free form being dialyzable. In general, dose reductions are recommended in addition to timing treatment immediately after HD sessions or on non-dialysis days [3,4]. Etoposide is partially (40–60%) excreted by the kidneys and is not removed by HD; dose reduction and treatment either before or after HD are recommended [4]. Additionally, successful dose escalation of cisplatin and etoposide administration with HD post infusion in lung cancer patients has been described [5].

In testicular cancer, however, the goal is cure, and any dose delays or reductions may impact cure rates. Due to the rare clinical situation in which one must deliver chemotherapy to testicular cancer patients on or requiring HD, there is no established published approach.

On a review of the literature at the time of planning treatment for our patient, one case report was found that described a patient with advanced nonseminomatous germ cell tumour on chronic HD with a known seizure history [6]. Cycle 1 consisted of cisplatin 50% dose and etoposide 60% dose with hemodialysis 1 h after infusion. Cycle 2 consisted of cisplatin and etoposide at full dose. With cycle 3, an ifosfamide 100% dose was added, but the patient developed pneumonia and bacteremia, requiring admission to Intensive Care. Although delayed, the fourth cycle consisted of cisplatin and etoposide at 100% dose and ifosfamide at 75% dose. The patient had recurrent seizures on day two of cycle 4 and another admission to the Intensive Care Unit; thus, no further ifosfamide was given.

Since treating our patient, two other English language case reports have been published. One documents a 28-year-old man with good risk nonseminomatous testicular cancer on HD treated with cisplatin 50% dose and etoposide 60% dose with HD every second day for cycle 1; however, the timing of HD is not described. The patient had anemia and thrombocytopenia [3]. Two subsequent cycles were given at full dose with daily HD during chemotherapy and G-CSF support. Post cycle 2, he developed thrombocytopenia, and post cycle 3, he developed pancytopenia but no clinical complications. He had a complete response. A second report documents a 31-year-old with end stage renal disease and a primary mediastinal yolk sac tumor. He was operated on first and then treated with cisplatin 50% dose and etoposide 60% dose with daily HD starting 1 h after chemotherapy and G-CSF support [7]. After the first cycle, he developed grade 4 neutropenia and appendicitis. Cycle 2 was given at the same dose with no complications, and thu, cycles

3–5 were given with cisplatin and etoposide 75% dose. He did develop significant pancytopenia but no further clinical complications. He had no cancer recurrence after one year. Cisplatin pharmacokinetic studies were available with cycles 3 and 4 which showed higher maximum drug concentrations and area under the curve despite dose reductions compared to expected pharmacokinetics in a patient with normal renal function. No excess non-hematological toxicities were reported in either of these case reports.

Despite guidelines and data supporting the use of three chemotherapy drugs in poor prognosis patients, neither bleomycin nor ifosfamide were used in our case. Bleomycin is predominantly cleared by the kidneys, is effectively not dialyzed, and can have life-threatening pulmonary toxicities, and thus it was felt to be too high risk to use in this situation [8]. Ifosfamide can be nephrotoxic but has been shown to be at least partially dialyzable [9]. If our patient had done well with cycle one, the plan had been to increase the etoposide dose first and then potentially add ifosfamide. This plan was abandoned due to the neutropenic complications seen after the first cycle despite G-CSF.

While five cycles of dose reduced chemotherapy produced a successful oncologic outcome, our patient suffered life altering long-term complications from sensory and motor neuropathy. Given he had grade 1 neuropathy after his fourth cycle, the fifth cycle of chemotherapy undoubtedly contributed to his morbidity.

Patients with metastatic testicular cancer on HD can be successfully treated with cisplatin and etoposide chemotherapy, although the optimal dose and schedule of the chemotherapy and HD remains unknown. Limited and conflicting data on chemotherapy drug concentrations, clearance, and availability in these situations make it difficult to make decisions on dose escalation or drug continuation. Should patients develop non-hematological toxicities during chemotherapy, one needs to weigh the risk–benefit ratio carefully between the risk of long-term complications and the potential impact on cure. This case report represents a significant contribution to the current literature on the safe delivery and efficacy for testicular cancer patients undergoing curative chemotherapy on HD, a topic for which further knowledge is imperative to guide future clinicians and establish future guidelines for this rare clinical scenario.

**Author Contributions:** Conceptualization, K.J.M. and L.A.W.; methodology, K.J.M., L.A.W. and S.S.; validation, K.J.M., L.A.W. and S.S.; writing—original draft preparation, K.J.M.; writing—review and editing, K.J.M., L.A.W. and S.S.; supervision, L.A.W. All authors have read and agreed to the published version of the manuscript.

**Funding:** This research received no external funding.

**Institutional Review Board Statement:** The study was conducted in accordance with the Declaration of Helsinki, and approved by the Nova Scotia Health Research Ethics Board Office (Approval code 1027810).

**Informed Consent Statement:** Informed consent was obtained from the subject involved in the study.

**Data Availability Statement:** The data presented in this study are available on request from the corresponding author.

**Conflicts of Interest:** The authors declare no conflict of interest.

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
