# Peer review of "Delivering Chemotherapy to a Metastatic Poor Risk Testicular Cancer Patient on Hemodialysis"

_curroncol, doi:10.3390/curroncol29030148_

Round 1
Reviewer 1 Report
This study evaluated the curative intend of chemotherapy for metastatic testicular cancer in patients undergoing hemodialysis. The study was well conducted the results are presented in a logical manner and the discussion flows well. The manuscript is well organized and written.
Delivering chemotherapy for metastatic testicular cancer in patients with renal function impairment is challenging and much more for patients in hemodialysis
I think the paper should be accepted in its present form without any revisions
The bibliographic reviews are old but this is due to the selection of the case reports. However bibliography 1 should be updated.
Author Response
Thank you for reviewing our report. We appreciate your positive comments. With regards to your comment and suggestion:
"The bibliographic reviews are old but this is due to the selection of the case reports. However bibliography 1 should be updated. "
We agree, some of the references are old which reinforces the rarity and lack of data on this topic. In terms of bibliography (or reference 1) from 2008, we included this reference as it is still a very thorough review of clinical trial data on chemotherapy options for advanced GCT. However, we appreciate your point with regards to including more contemporary references, and thus have changed this reference to the European Association of Urology Testicular Cancer Guidelines which are an excellent resource and always up to date.
Reviewer 2 Report
Dr. Moore et al., presented a case report on treatment approach for chemotherapy treatment in testicular cancer patients on hemodialysis. They presented a rationale for treating testicular cancer patients on hemodialysis with optimal chemotherapeutic drug doses and timing theraphy with hemodialysis to achieve maximum curative benefits while minimizing toxicity.
The authors have presented clear and helpful insights into effective and safe delivery of chemotherapeutics in these subset of patients.
The treatment plan considered by the authors is based on keen knowledge on metabolism of chemotherapeutic agents and risk vs benefit, which illuminates a path forward to follow for future treatments in these subset of patients.
The authors have contributed considerably to the very little literature available for delivering optimal doses of chemotherapeutic drugs to these patients. I highly commend the efforts of the authors and for the insights they provided.
Author Response
Thank you so much for your positive comments. We do hope this case report helps clinicians facing this challenge in the future.
Reviewer 3 Report
The Authors present a rare case of high-risk, metastatic NSGCT in an ESRD patient, successfully treated with orchidectomy, five cycles of dose-reduced EP chemotherapy with synchronous hemodialysis, and a liver metastasectomy.
Although there were several case reports on use of cisplatin-based chemotherapy in HD patients with GCTs, I believe this case report contributes significantly to the scarce evidence in this area. The manuscript is well-written, comprehensive and provides long-term follow-up data.
My remarks are following:
- The authors describe findings of laboratory tests, ultrasound, CT-scan, bone scan, only to mention findings of physical examination showing a rather obvious 16cm(!) testicular lesion towards the end. I would advise to rearrange the introduction to represent appropriate sequence of patient work-up (I suppose the PE was done as an initial Pt workup)
- Please improve keywords,
- Description of Figure 2 refers to liver metastasis, whilst the figure itself shows a scrotal mass.
- The authors refer to etoposide (60mg/m2) as being 60% dose, would be good to stress that cisplatin was administered at a full dose (20mg/m2), despite ESRD.
- I presume the bone scan lesions were followed? Have they regressed or described nonmalignant in subsequent FU?
- Has the periportal lymphadenopathy met criteria for pathologic enlargement on CT? If so, the patient shall be considered cN+.
Author Response
Thank you for your very through and comprehensive review. Your comments are greatly appreciated. We will address them individually below.
- Sadly, the sequence of the investigations as written in the report is indeed the order in which things occurred. He presented with abdominal pain which led to bloodwork, which led to an ultrasound, which led to a CT scan, which showed a testicular mass and then an exam. So, we decided to report it as the events unfolded. As you may know, it is not uncommon that men do not offer a history of a testicular mass and/or health care professionals do not inquire about testicular masses or examine the testicles initially. If you would still like us to move up the physical exam of the testicles, we certainly can.
- Improve the key words. Thank you, we have done so and hope they meet with Current Oncology's approval as a few are two words (Chemotherapy delivery, hemodialysis, renal failure)
- Mislabeling Figure 2. Thank you for pointing this out. We are sorry for this oversight. It has been corrected. (line 160)
- Emphasize that Cisplatin 20 mg/m2 is full dose. Thank you. We have added in the words "full dose" on line 56 before "Cisplatin 20 mg/m2".
- Comment on the status of the bone lesions. Thank you for asking about this. We have added a sentence in line 92 that indicates "the bone lesions regressed over time".
- Clarify whether periportal lymph nodes should categorize the patient as cN+. We don't consider the perioportal lymph nodes as regional lymph nodes but instead non regional lymph nodes which would be considered M1a disease (definition of M1a = N-regional lymph node(s) or lung metastases).